# Predictive Performance of Scoring Systems for Mortality Risk in Patients with Cryptococcemia: An Observational Study

**DOI:** 10.3390/jpm13091358

**Published:** 2023-09-06

**Authors:** Wei-Kai Liao, Ming-Shun Hsieh, Sung-Yuan Hu, Shih-Che Huang, Che-An Tsai, Yan-Zin Chang, Yi-Chun Tsai

**Affiliations:** 1Institute of Medicine, Chung Shan Medical University, Taichung 40201, Taiwan; kents90124@hotmail.com; 2School of Medicine, Chung Shan Medical University, Taichung 40201, Taiwan; cucu0214@gmail.com; 3Department of Emergency Medicine, Taichung Veterans General Hospital, Taichung 407219, Taiwan; rosa87324@gmail.com; 4Department of Post-Baccalaureate Medicine, College of Medicine, National Chung Hsing University, Taichung 402, Taiwan; 5School of Medicine, National Cheng Kung University, Tainan 701, Taiwan; 6School of Medicine, National Yang Ming Chiao Tung University, Taipei 11217, Taiwan; edmingshun@gmail.com; 7Department of Emergency Medicine, Taipei Veterans General Hospital, Taoyuan Branch, Taoyuan 330, Taiwan; 8Department of Emergency Medicine, Taipei Veterans General Hospital, Taipei 11217, Taiwan; 9Department of Emergency Medicine, Chung Shan Medical University Hospital, Taichung 40201, Taiwan; 10Lung Cancer Research Center, Chung Shan Medical University Hospital, Taichung 40201, Taiwan; 11Division of Infectious Disease, Department of Internal Medicine, Taichung Veterans General Hospital, Taichung 40705, Taiwan; lucky-sam@yahoo.com.tw; 12Department of Clinical Laboratory, Drug Testing Center, Chung Shan Medical University Hospital, Taichung 40201, Taiwan

**Keywords:** cryptococcus, emergency department, mortality risk, risk factors, scoring systems

## Abstract

Cryptococcal infection is usually diagnosed in immunocompromised individuals and those with meningeal involvement, accounting for most cryptococcosis. Cryptococcemia indicates a poor prognosis and prolongs the course of treatment. We use the scoring systems to predict the mortality risk of cryptococcal fungemia. This was a single hospital-based retrospective study on patients diagnosed with cryptococcal fungemia confirmed by at least one blood culture collected from the emergency department covering January 2012 and December 2020 from electronic medical records in the Taichung Veterans General Hospital. We enrolled 42 patients, including 28 (66.7%) males and 14 (33.3%) females with a mean age of 63.0 ± 19.7 years. The hospital stay ranged from 1 to 170 days (a mean stay of 44.4 days), and the overall mortality rate was 64.3% (27/42). In univariate analysis, the AUC of ROC for MEWS, RAPS, qSOFA, MEWS plus GCS, REMS, NEWS, and MEDS showed 0.833, 0.842, 0.848, 0.846, 0.846, 0.878, and 0.905. In the multivariate Cox regression analysis, all scoring systems, older age, lactate, MAP, and DBP, indicated significant differences between survivor and non-survivor groups. Our results show that all scoring systems could apply in predicting the outcome of patients with cryptococcal fungemia, and the MEDS displays the best performance. We recommend a further large-scale prospective study for patients with cryptococcal fungemia.

## 1. Introduction

Cryptococcus has encapsulated yeast that lives in the natural environment, but it is rarely a pathogen in individuals with a healthy immune system. In literature reviews, *Cryptococcus neoformans* is humans’ most common pathogenic cryptococcal species, usually diagnosed in immunocompromised individuals [1]. In addition, *Cryptococcus gattii* is reported as a rare pathogen in cryptococcosis cases and is predominantly a causative pathogen in immunocompetent individuals [2,3]. The central nervous system (CNS) involvement generally accounts for most cryptococcosis [4], so the clinicians suggest evaluating CNS involvement in patients with evidence of cryptococcal infection. Cryptococcal meningitis is estimated to be associated with human immunodeficiency virus (HIV), with the global occurrence of 223,100 cases in 2014 and the annual deaths of 181,100 patients [5]. Other possible infected sites of cryptococcosis, including the respiratory tract, urinary tract, skin, bone, eye, and gastrointestinal tract, have been reported in previous studies [6,7,8,9,10,11,12,13,14,15].

Cryptococcemia occurred in only 10% to 30% of all cryptococcal diseases but was often associated with prolonging the clinical course of treatment and higher mortality rates [16,17,18,19,20]. Unfortunately, few published articles on cryptococcemia analyzed the clinical characteristics and outcomes according to clinical presentations, comorbidities, and scoring systems. Previous studies reported the presence of an immunocompromised condition, liver cirrhosis, high Acute Physiology and Chronic Health Evaluation (APACHE) II score (≥20), and severity of sepsis, and they were associated with a higher mortality rate [18,20]. Published articles did not establish the predictive factors or scoring systems to evaluate cryptococcemia.

In recent studies, they applied various simple scoring systems (Appendix A), including quick the Sequential Organ Failure Assessment (qSOFA) Score, Rapid Acute Physiology Score (RPAS), Mortality in Emergency Department Sepsis (MEDS) score, Modified Early Warning Score (MEWS), National Early Warning Score (NEWS), and Rapid Emergency Medicine Score (REMS), to become the predictors of clinical outcomes for critical illness. However, they did not apply these in the survey of cryptococcemia [21,22,23,24,25,26,27,28]. Therefore, we analyzed the risk factors for patients with cryptococcemia and the impact of mortality rate by different origins of cryptococcal infection, clinical characteristics, and the performance of the abovementioned scoring systems.

## 2. Materials and Methods

### 2.1. Data Collection and Definition

The institutional review board of Taichung Veterans General Hospital (TCVGH), Taichung, Taiwan, approved our study (CE22240B). It was a single hospital-based retrospective study on patients with cryptococcal fungemia confirmed by at least one blood culture collected from the emergency department (ED) [29]. We excluded patients only presenting the positive cryptococcal antigen without the growth of cryptococcus in blood culture.Patients’ data, including clinical characteristics, comorbidities, laboratory investigations, co-infection conditions, hospital course, and mortality rate, were collected between January 2012 and December 2020 from the electronic medical records (EMRs) in TCVGH. We collected patients’ vital signs and laboratory data to analyze scoring systems during blood culture, which identified cryptococcal fungemia. The primary outcome was the overall in-hospital mortality rate. We excluded patients younger than 18 years old or transferring to other hospitals. We defined cryptococcosis as a positive culture of *Cryptococcus neoformans* yielded from the various specimens of the clinically involved sites, including cerebrospinal fluid (CSF), sputum/bronchial lavage, urine, ascites, and skin biopsy.We collected blood cultures and vital signs for analyses in the case of identified cryptococcal fungemia. We defined septic shock as needing inotropic agents or vasopressors to correct hypotension and lactic acidosis resulting from infection.

### 2.2. Scoring Systems

We collected all of the parameters for analysis in the scoring systems from the EMRs. The clinical scoring systems of this study included qSOFA, RAPS, MEDS, MEWS, NEWS, and REMS.

### 2.3. Statistical Analysis

We presented continuous data as mean ± standard deviation (SD). We expressed categorical data as numbers and percentages. Chi-squared tests were applied to compare categorical data. Mann–Whitney–Wilcoxon U-tests were involved to compare continuous data regarding mortality risks in survivors and non-survivors. To assess possible predictors for mortality, we conducted univariate and multivariate analyses using the Cox regression model to express results as confidence interval and hazard ratio. We used the area under the curve (AUC) receiver operating of the characteristic curve (ROC) to compare predictive power across different scoring systems. We used cut-off points of scores to stratify mortality risks in terms of sensitivity, specificity, negative predictive value (NPV), and positive predictive value (PPV). A *p* value < 0.05 was considered statistically significant. We analyzed the data using the Statistical Package for the Social Science (IBM SPSS version 22.0; International Business Machines Corp., New York, NY, USA) and R (Version 4.1.3, R Foundation for Statistical Computing, Vienna, Austria).

## 3. Results

### 3.1. Demographics and Clinical Characteristics

We identified 43 patients with cryptococcemia from January 2012 to December 2020, and excluded one patient due to transferring to other hospitals before being diagnosed with cryptococcal fungemia. Finally, we enrolled 42 patients in our study. There were 28 (66.7%) males and 14 (33.3%) females with a mean age of 63.0 ± 19.7 years. The total hospital stays ranged from 1 to 170 days (a mean stay of 44.4 days), and the overall mortality rate was 64.3% (27/42). Only one patient did not have immunodeficient status. Of the remaining 41 patients, there were 16 under immunosuppressants (such as steroids or immunomodulatory drugs for autoimmune disorders or organ transplants), 8 with HIV infection, 6 with liver cirrhosis, 6 with diabetes mellitus (DM), 5 with end-stage renal disease (ESRD), and 4 under chemotherapy due to neoplasms or hematologic disorders. Among all the comorbidities, the prevalence of HIV infection was higher in the survivors than in the non-survivors (40.0% vs. 7.4%, *p* = 0.016). We summarized the demographics and clinical characteristics, laboratory data, and scoring systems of 42 patients in Table 1. In the subgroup analysis of 30 patients who underwent lumbar puncture, we concluded their characteristics and laboratory investigations in Table 2.

We showed patient distribution in different seasons and the average temperature of each season. There was an increasing overall patient number according to the higher average temperature in different seasons (*p* = 0.044). Moreover, increased cases of mortalitywere also associated with increased average temperature (*p* = 0.014) and low average temperature (*p* = 0.030) in the different seasons (Figure 1).

### 3.2. Laboratory Data and Scoring Systems

We showed laboratory data and scoring systems in Table 1. The non-survivors had lower hemoglobin (Hb) (11.0 ± 1.6 vs. 9.5 ± 2.6, *p* = 0.021), lower platelet (PLT) counts (244.7 ± 124.5 vs. 128.7 ± 94.9, *p* = 0.002), and a higher level of creatinine (1.0 ± 0.4 vs. 2.6 ± 2.5, *p* = 0.003) than the survivors. In addition, all scoring systems showed significantly higher scores in the non-survivors than in the survivors.

### 3.3. Microbiology

Our study identified no other cryptococcal species but *Cryptococcus neoformans* in patients with cryptococcal fungemia.We suggested that all patients undergo an examination of CSF once the diagnosis of cryptococcemia was confirmed. However, only 22 patients confirmed diagnosis of cryptococcosis with CNS involvement. Additionally, *Cryptococcus neoformans* was isolated from urine in four patients, the respiratory tract in two, skin biopsy in one, and peritoneal fluid in one. We confirmed 12 patients with a diagnosis of primary cryptococcemia. Another 12 patients, assumed as primary cryptococcemia, passed away before undergoing an examination of CSF or declining lumbar puncture.

### 3.4. Clinical Outcomes and Co-Infections Related to Other Pathogens

Twenty-seven patients passed away during hospitalization, with an overall in-hospital mortality rate of 64.3%. We found co-infections of urinary tract infections in 8 patients, pneumonia in 16, and primary bacteremia in 21.

### 3.5. Univariate and Multivariate Analysis of Risk Factors

In the univariate analysis, older age, no CNS involvement, female gender, high respiratory rate, oxygen (O_2_) use, low scores of Glasgow Coma Scale (GCS), combined lower respiratory tract infection (LRTI), elevated white blood cell (WBC) counts, low PLT counts, high levels of lactate and creatinine, high scores of qSOFA, RAPS, MEDS, MEWS, MEWS GCS, REMS, and NEWS were associated with a higher overall in-hospital mortality rate (Table 3). In the multivariate analysis, MEDS presented a higher in-hospital mortality rate (HR: 1.21, 95% CI: 1.03–1.41, *p* = 0.018) (Table 4).

### 3.6. Receiver Operating Characteristic Curve (ROC)

The AUC of ROC for MEWS, RAPS, qSOFA, MEWS plus GCS, REMS, NEWS, and MEDS showed 0.833, 0.842, 0.848, 0.846, 0.846, 0.878, and 0.905, respectively. They performed well in predicting the in-hospital mortality risk of patients with cryptococcal fungemia. The MEDS showed the best performance in predicting the mortality risk, and the AUC of ROC was 0.905 at the cut-off points of 4 in Figure 2. The sensitivity and specificity of the MEDS in predicting the mortality risk were 93% and 80% (Table 5).

### 3.7. Cumulative Survival Rates Using Kaplan–Meier and Discrimination Plots

We calculated the cumulative survival rates of patients with cryptococcemia to predict the 30-day mortality rate using Kaplan–Meier analyses (Figure 3). The cut-off points of MEDS, NEWS, qSOFA, MEWS plus GCS, REMS, RAPS, and MEWS were 4, 5, 1, 3, 8, 3, and 3, respectively. Furthermore, the overall mortality case numbers of MEDS, NEWS, qSOFA, MEWS plus GCS, REMS, RAPS, and MEWS were 25, 22, 13, 20, 14, 15, and 16, with the overall mortality rate of 89.3%, 91.7%, 92.9%, 87.0%, 100%, 100%, and 88.9% if the cut-off points were more than 4, 5, 1, 3, 8, 3, and 3, respectively, which is shown in the discrimination plots in Figure 4.

## 4. Discussion

In the literature, this is the first study of applying scoring systems in predicting the mortality risk of patients with cryptococcemia and identifying higher scores associated with a significantly higher mortality rate in patients with cryptococcemia.

In this study, the overall in-hospital mortality rate of cryptococcemia was 64.3% (27/42), and the 30-day mortality rate was 47.6% (20/42), which was higher compared to the previously reported 30-day mortality rate of 35% [18,19,20]. This may be related to higher age, with a mean age of 63 years, compared to a mean age of 40–50 years in previous studies [18,19,20]. We also found a higher incidence of primary cryptococcemia (12/42, 28.6%), which may be related to a higher incidence of patients’ mortality. Meningeal involvement could account for most cryptococcosis. The higher incidence of cryptococcosis made physicians aware of cryptococcal infections in the HIV population. Hence, all patients with HIV infection received an examination for CSF during hospitalization in our study. Of all the 30 patients who received an analysis for CSF, 22 patients were diagnosed with cryptococcal meningitis through a positive culture of CSF or positive findings of India ink. We found 73.3% CNS involvement in 30 patients with cryptococcemia, similar to the previous report of 71.0–89.3% with CNS involvement in patients with cryptococcemia [18,20].

*Cryptococcus neoformans* distribute worldwide, and the optimal temperature of *Cryptococcus neoformans* is 30 °C in laboratory conditions [29]. The publishedliterature report showed a relatively higher growth rate in the planktonic type at 30 °C compared to 35 °C, but better biofilm growth at 35 °C [30]. A systemic investigation in Colombia showed *Cryptococcus neoformans* to be more easily isolated in cold temperate climates and related to higher humidity or lower sunshine [31]. In our research, we found an increasing trend inthe prevalence of cryptococcemia in the summer, and the average summer temperature in Taiwan was also closest to 30 °C. Additionally, we also found the tendency of a positive relationship between mortality case numbers and both high and low average temperatures. However, there was no significant difference in mortality rate in different seasons, which may be related to the few case numbers. Therefore, the possibility of a positive relationship between temperature and cryptococcal infection is worthy of further investigation.

The risk factors of mortality in cryptococcal fungemia of this study were higher age, being female, no CNS involvement, higher respiratory rate, O_2_ use, lower GCS level, combined LRTI, higher WBC counts, lower PLT counts, more elevated lactate and creatinine levels, and higher scores of qSOFA, RAPS, MEDS, MEWS, MEWS GCS, REMS, and NEWS. In addition, we found patients with CNS involvement with a lower mortality rate in the univariate analysis of all populations but a non-significant difference in the subgroup analysis of 30 patients who underwent an examination of CSF (Table 2). We supposed the possibility that patients who died before undergoing a test of CSF were classified into the non-CNS involvement group, resulting in a higher mortality rate of the non-CNS involvement group.

Generally, amphotericin B, liposomal amphotericin, and azoles (including fluconazole and itraconazole) were considered effective antifungal agents for cryptococcosis [20,32]. However, nine patients passed awaywithout being prescribed effective antifungal agents, and only one patient receiving amphotericin B passed awaywithin 24 h in our study. Although the abovementioned patients were diagnosed with cryptococcemia by blood culture after mortality, all received amphotericin B according to the positive cryptococcal antigen in the examination for CSF.

These clinical scoring systems have been a clinical tool applied to evaluate the mortality risk in the ED or general ward. The qSOFA was first created in 2016 as a measuring tool to determine critical conditions in septic patients [21]. Still, it was considered to have a lower prognostic accuracy for in-hospital mortality or risk of intensive care unit (ICU) admission than the SOFA score [22]. The RAPS, as an abbreviated version of the APECHE-II score, was developed in 1987 as a severity scale in critical care transport and was considered for its predictive ability regarding mortality risk by only using simple parameters available on transported patients. The RAPS is further applied to evaluate the mortality risk of the patients in the ED or other different categories of patients [23]. The MEWS was investigated in 2001 and was created to identify patients in busy areas with a risk of deterioration. A reported score of 5 or more was associated with a higher risk of death or depravation [24]. The NEWS, created by teams at the Royal College of Physicians in London, was recommended as an early discriminating tool for patients at risk of cardiac arrest, with unplanned admission to the ICU, or in the case of death within 24 h with a high AUROC [25]. In 2004, the REMS extended the RAPS by adding the patient’s age and peripheral oxygen saturation to predict the in-hospital mortality of the non-surgical ED patients.

Further investigation demonstrated its superior predictive value in comparison to the RAPS [26]. Shapiro et al. developed the MEDS score initially according to the odds ratio of mortality risk in ED patients with a chance of sepsis in 2003. In Taiwan, many authors applied the MEDS score to predict the severity and mortality rate of patients with bacteremia [27,28,33].

In our study, higher scores in all the abovementioned scoring systems were associated with a substantially higher mortality rate, and the MEDS presented the best predictive performance. In general, the MEDS is composed of clinical manifestations and laboratory data, including age >65 years (3 points), nursing home residence (2 points), terminal illness (6 points), altered mental status (2 points), tachypnea or hypoxia (3 points), septic shock (3 points), LRTI (2 points), PLT counts <150,000 (3 points), and band portion >5% (3 points) with a maximum of 27 points. The AUC of ROC of MEDS was 0.905 at the cut-off point of 4, with a sensitivity of 92.6% (25/27) and specificity of 80% (12/15), respectively. However, we found a lower cut-off point than other studies [27,28,34]. The possible causes were the presence of no patients with nursing home residence or terminal illness conditions and no band portion >5% in our study.

## 5. Limitations

First, our study had some limitations due to the retrospective nature and small sample size. For example, the possible time lag between collecting the first blood culture to identify cryptococcemia and the records of vital signs or laboratory data—or missing data, presented in some cases—such as relating to lumbar puncture. Second, antifungal agents or other management procedures, after diagnosis of cryptococcemia, were not standardized, so we cannot compare the clinical outcomes according to those treatments. Third, the enrolled patients had high rates of comorbidities and co-infections. Fourth, cryptococcemia is a rare disease. Therefore, there is a lack of awareness in patients initially presented in the ED.Also, the long inoculation time of cryptococcus species’ growth makes it difficult to conduct a prospective study and to standardize the treatment protocol for patients with cryptococcemia.

## 6. Conclusions

Cryptococcemia is a rare entity, but it is life-threatening with a high mortality rate, so physicians should maintain suspicion in high-risk patients. An examination for CSF in patients with cryptococcemia is strongly recommended, due to the fact that the majority of patients with cryptococcemia present CNS involvement. Age, gender, respiratory rate, O_2_ use, GCS level, LRTI, WBC counts, PLT counts, lactate, creatinine, and high points of scoring systems are associated with poor prognosis in patients with cryptococcemia. The MEDS (≥4) performs best in predicting mortality risk. We recommend further large-scale studies on early detection through the biomarkers of cryptococcemia and the appropriate use of scoring systems to predict the mortality risk in order to improve clinical outcomes.

## Figures and Tables

**Figure 1 jpm-13-01358-f001:**
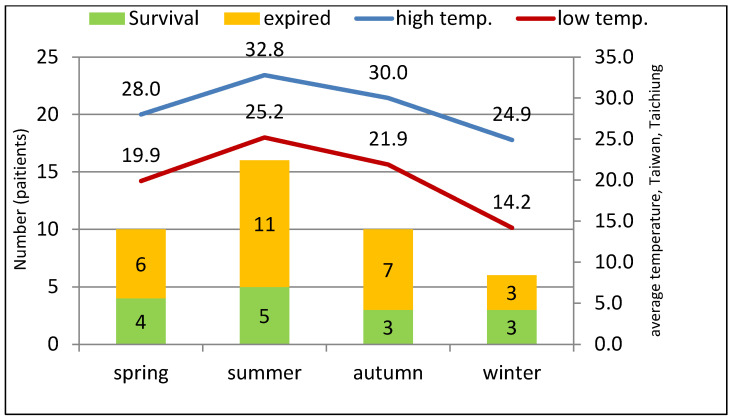
The trend association between the seasonal temperature and total deceased patient numbers of cryptococcemia. Trend of high temperature and total patients, *p* = 0.044. Trend of high temperature and deceased patients, *p* = 0.014; Trend of low temperature and deceased patients, *p* = 0.030.

**Figure 2 jpm-13-01358-f002:**
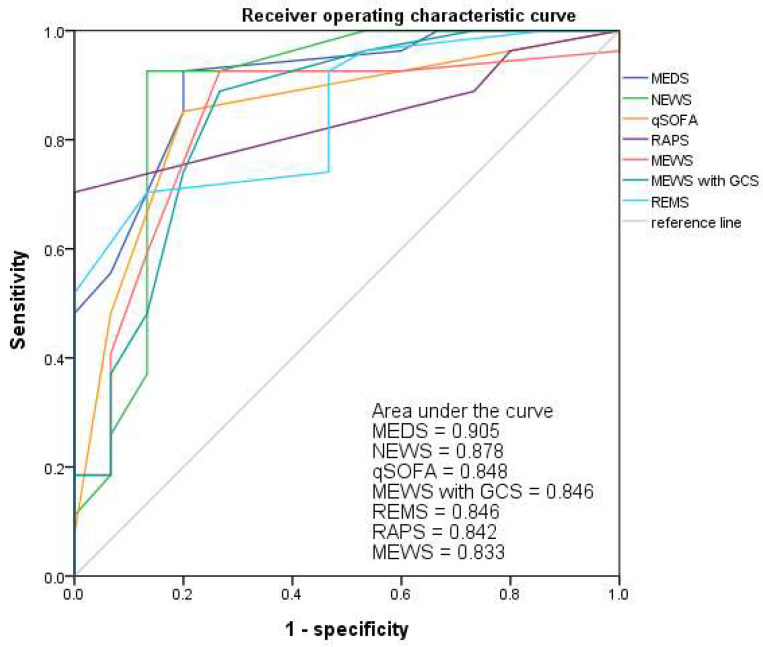
The AUC of ROC for MEDS, NEWS, qSOFA, MEWS plus GCS, REMS, RAPS, and MEWS indicated 0.905, 0.878, 0.848, 0.846, 0.846, 0.842, and 0.833 to predict the mortality risks of patients with cryptococcemia. AUC, area under the curve; GCS, Glasgow coma scale; MEDS, Mortality in Emergency Department Sepsis Score; MEWS, Modified Early Warning Score; NEWS, National Early Warning Score; qSOFA, quick Sequential Organ Failure Assessment; RAPS, Rapid Acute Physiology Score; REMS, Rapid Emergency Medicine Score; ROC, receiver operating characteristic curve.

**Figure 3 jpm-13-01358-f003:**
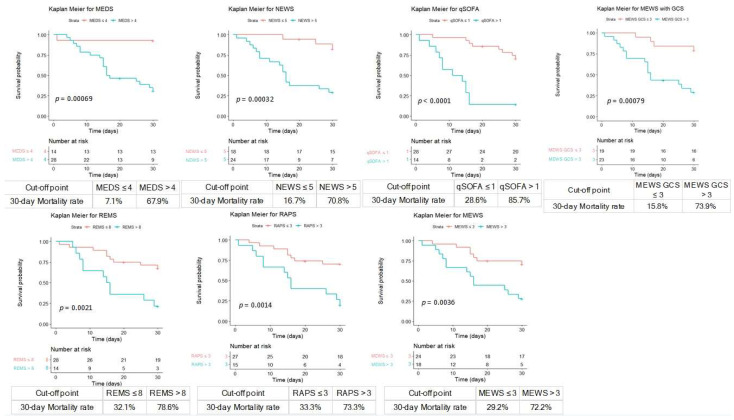
The cumulative survival rates of patients with cryptococcemia were calculated to predict the 30-day mortality rate using Kaplan–Meier analyses. The cut-off point of MEDS, NEWS, qSOFA, MEWS plus GCS, REMS, RAPS, and MEWS was 4, 5, 1, 3, 8, 3, and 3, respectively. GCS, Glasgow coma scale; MEDS, Mortality in Emergency Department Sepsis Score; MEWS, Modified Early Warning Score; NEWS, National Early Warning Score; qSOFA, quick Sequential Organ Failure Assessment; RAPS, Rapid Acute Physiology Score; REMS, Rapid Emergency Medicine Score.

**Figure 4 jpm-13-01358-f004:**
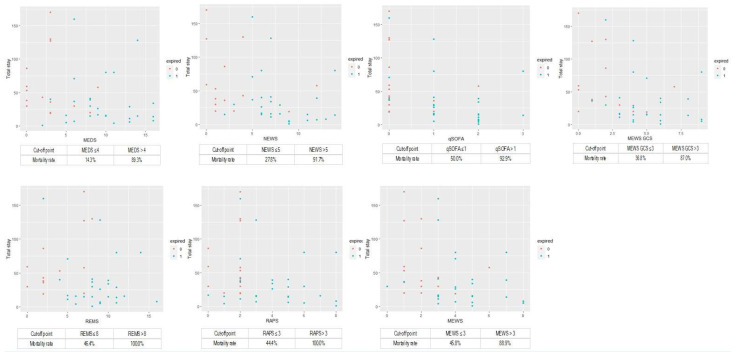
The overall mortality case numbers of MEDS, NEWS, qSOFA, MEWS plus GCS, REMS, RAPS, and MEWS were 25, 22, 13, 20, 14, 15, and 16, with the overall mortality rate of 89.3%, 91.7%, 92.9%, 87.0%, 100%, 100%, and 88.9% if the cut-off point was more than 4, 5, 1, 3, 8, 3, and 3, respectively.GCS, Glasgow coma scale; MEDS, Mortality in Emergency Department Sepsis Score; MEWS, Modified Early Warning Score; NEWS, National Early Warning Score; qSOFA, quick Sequential Organ Failure Assessment; RAPS, Rapid Acute Physiology Score; REMS, Rapid Emergency Medicine Score.

**Table 1 jpm-13-01358-t001:** Demographics and laboratory data of 42 patients with fungemia of *Cryptococcus neoformans*.

General Data	Patients (*n* = 42)	Survivors (*n* = 15)	Non-Survivors (*n* = 27)	*p*-Value
Age (years)	63.0 ± 19.7	52.5 ± 19.7	68.9 ± 17.4	0.014 *
Male (%)	28 (66.7%)	12 (80%)	16 (59.3%)	0.172
Hospital stays (days)	44.4 ± 42.9	61.3 ± 46.5	35.0 ± 38.5	0.008 **
Focus of cryptococcosis				
CNS ^f^	22 (52.4%)	11 (73.3%)	11 (40.7%)	0.009 **
Respiratory tract ^f^	4 (9.5%)	2 (13.3%)	2 (7.4%)	0.608
Clinical conditions				
Septic shock ^f^	7 (16.7%)	0 (0.0%)	7 (25.9%)	0.038 *
IICP ^f^	18 (42.9%)	8 (53.3%)	10 (37.0%)	0.007 **
Concomitant infections				
Pneumonia	16 (38.1%)	5 (33.3%)	11 (40.7%)	0.746
Urinary tract ^f^	8 (19.1%)	2 (13.3%)	6 (22.2%)	0.689
Bacteremia	21 (50.0%)	7 (46.7%)	14 (51.9%)	1
Comorbidities				
HIV ^f^	8 (19.1%)	6 (40.0%)	2 (7.4%)	0.016 *
Liver cirrhosis ^f^	6 (14.3%)	1 (6.7%)	5 (18.5%)	0.395
ESRD ^f^	5 (11.9%)	0 (0.0%)	5 (18.5%)	0.142
DM ^f^	6 (14.3%)	1 (6.7%)	5 (18.5%)	0.395
Immunosuppressant use	16 (38.1%)	5 (33.3%)	11 (40.7%)	0.746
Under chemotherapy ^f^	4 (9.5%)	1 (6.7%)	3 (11.1%)	1
Vital signs				
SBP (mmHg)	134.6 ± 26.8	136.8 ± 20.1	133.4 ± 30.2	0.763
DBP (mmHg)	80.6 ± 19.4	79.9 ± 12.6	81.0 ± 22.5	0.937
MAP (mmHg)	98.6 ± 20.5	98.8 ± 13.5	98.4 ± 23.7	1
HR (bpm)	97.4 ± 25.5	96.2 ± 18.5	98.0 ± 29.0	0.636
RR (bpm)	20.1 ± 3.7	18.5 ± 1.6	20.9 ± 4.3	0.022 *
BT (°C)	37.7 ± 0.9	37.7 ± 1.2	37.7 ± 0.8	0.590
SpO_2_ (%)	95.6 ± 5.4	96.1 ± 2.6	95.3 ± 6.5	0.355
O_2_ use	26 (61.9%)	3 (20.0%)	23 (85.2%)	<0.001 **
GCS	11.6 ± 4.2	14.6 ± 1.1	10.0 ± 4.4	<0.001 **
Laboratory data				
WBC (counts/uL)	9312.9 ± 6707.0	7026.0 ± 5071.7	10,583.3 ± 7238.2	0.125
Hb (g/dL)	10.1 ± 2.4	11.0 ± 1.6	9.5 ± 2.6	0.021 *
PLT (×10^3^ counts/uL)	170.2 ± 119.0	244.7 ± 124.5	128.7 ± 94.9	0.002 **
Crea (mg/dL)	1.97 ± 2.17	0.96 ± 0.42	2.56 ± 2.54	0.003 **
Lactate (mg/dL)	22.1 ± 28.6	11.6 ± 4.6	25.6 ± 32.3	0.104
pH	7.40 ± 0.07	7.42 ± 0.04	7.39 ± 0.07	0.427
Scoring systems				
qSOFA	1.0 ± 0.9	0.3 ± 0.6	1.4 ± 0.8	<0.001 **
RAPS	3.2 ± 2.2	1.5 ± 0.8	4.1 ± 2.3	<0.001 **
MEWS	3.5 ± 2.0	2.1 ± 1.4	4.3 ± 1.9	<0.001 **
MEWS with GCS	3.9 ± 2.6	2.0 ± 2.0	4.9 ± 2.2	<0.001 **
REMS	7.1 ± 3.7	4.3 ± 3.0	8.6 ± 3.1	<0.001 **
NEWS	6.1 ± 4.1	2.9 ± 3.4	8.0 ± 3.2	<0.001 **
MEDS	7.1 ± 4.8	2.9 ± 2.9	9.5 ± 4.0	<0.001 **

Chi–squared test. ^f^ Fisher’s exact test. Mann–Whitney U-test.* *p* <0.05, ** *p* <0.01, Statistically significant. Continuous data were expressed as mean ± SD. Categorical data were expressed as number and percentage.BT, body temperature; CNS, central nervous system; Crea, Creatinine; DBP, diastolic blood pressure; DM, Diabetes Mellitus; ESRD, end–stage renal disease; GCS, Glasgow coma scale; HR, heart rate; Hb, hemoglobin; IICP, increased intracranial pressure; MAP, mean blood pressure; MEDS, Mortality in Emergency Department Sepsis Score; MEWS, Modified Early Warning Score; NEWS, National Early Warning Score; PLT, platelet; qSOFA, quick Sequential Organ Failure Assessment; RAPS, Rapid Acute Physiology Score; REMS, Rapid Emergency Medicine Score; RR, respiratory rate; SBP, systolic blood pressure; WBC, white blood cells.

**Table 2 jpm-13-01358-t002:** Demographics and laboratory data of 30 patients who underwent lumbar puncture and CSF examination.

General Data	Patients (*n* = 30)	Survivors (*n* = 15)	Non-Survivors (*n* = 15)	*p*-Value
Age (years)	57.5 ± 19.5	52.5 ± 19.7	62.6 ± 18.7	0.106
Male (%)	23 (76.7%)	12 (80.0%)	11 (73.3%)	1
Hospital stays (days)	53.5 ± 45.9	61.3 ± 46.5	45.7 ± 45.5	0.116
Focus of cryptococcosis				
CNS	22 (73.3%)	11 (73.3%)	11 (73.3%)	1
Respiratory tract	4 (13.3%)	2 (13.3%)	2 (13.3%)	1
Clinical conditions				
Septic shock	2 (6.7%)	0 (0.0%)	2 (13.3%)	0.483
IICP	18 (60.0%)	8 (53.3%)	10 (66.7%)	0.709
Concomitant infections				
Pneumonia	13 (38.1%)	5 (33.3%)	8 (53.3%)	0.461
Urinary tract	5 (16.7%)	2 (13.3%)	3 (20.0%)	1
Bacteremia	16 (53.3%)	7 (46.7%)	9 (60.0%)	0.714
Comorbidities				
HIV	8 (26.7%)	6 (40.0%)	2 (13.3%)	0.215
Liver cirrhosis	4 (13.3%)	1 (6.7%)	3 (20.0%)	0.598
ESRD	2 (6.7%)	0 (0.0%)	2 (13.3%)	0.483
DM	4 (13.3%)	1 (6.7%)	3 (20.0%)	0.598
Immunosuppressant use	12 (40.0%)	5 (33.3%)	7 (46.7%)	0.709
Under chemotherapy	2 (6.6%)	1 (6.7%)	1 (6.7%)	1
Vital signs				
SBP (mmHg)	139.2 ± 21.8	136.8 ± 20.1	141.5 ± 23.9	0.567
DBP (mmHg)	84.4 ± 17.9	79.9 ± 12.6	88.87 ± 21.4	0.217
MAP (mmHg)	102.6 ± 17.6	98.8 ± 13.5	106.4 ± 20.6	0.267
HR (bpm)	93.9 ± 24.3	96.2 ± 18.5	91.5 ± 29.5	0.744
RR (bpm)	19.0 ± 2.6	18.5 ± 1.6	19.4 ± 3.3	0.187
BT (°C)	37.7 ± 1.1	37.7 ± 1.2	37.8 ± 1.0	0.870
SpO_2_ (%)	96.2 ± 2.8	96.1 ± 2.6	96.3 ± 3.0	0.838
O_2_ use	14 (46.7%)	3 (20.0%)	11 (73.3%)	<0.001 **
GCS	12.6 ± 4.0	14.6 ± 1.1	10.5 ± 4.7	0.019 *
Laboratory data				
WBC (counts/uL)	7574.0 ± 4616.7	7026.0 ± 5071.7	8122.0 ± 4217.2	0.412
Hb (g/dL)	10.4 ± 2.5	11.0 ± 1.6	9.7 ± 3.0	0.098
PLT (×10^3^ counts/uL)	181.3 ± 124.3	244.7 ± 124.5	117.9 ± 88.8	0.002
Crea (mg/dL)	1.63 ± 2.10	0.96 ± 0.42	2.34 ± 2.87	0.026 *
Lactate (mg/dL)	14.1 ± 8.8	11.6 ± 4.6	15.6 ± 10.5	0.313
pH	7.42 ± 0.04	7.42 ± 0.04	7.41 ± 0.05	0.681
Scoring systems				
qSOFA	0.6 ± 0.8	0.3 ± 0.6	1.0 ± 0.8	0.011 *
RAPS	2.7 ± 2.1	1.5 ± 0.8	3.9 ± 2.3	0.002 **
MEWS	2.8 ± 1.6	2.1 ± 1.4	3.4 ± 1.5	0.013 *
MEWS GCS	3.0 ± 2.1	2.0 ± 2.0	4.1 ± 1.6	0.004 **
REMS	6.0 ± 3.3	4.3 ± 3.0	7.6 ± 2.8	0.005 **
NEWS	4.4 ± 3.1	2.9 ± 3.4	5.9 ± 1.8	0.001 **
MEDS	5.5 ± 4.3	2.9 ± 2.9	8.2 ± 3.8	<0.001 **

Chi–squared test. Mann–Whitney U-test. * *p* < 0.05, ** *p* < 0.01, Statistically significant. BT, body temperature; CNS, central nervous system; Crea, Creatinine; DBP, diastolic blood pressure; DM, Diabetes Mellitus; ESRD, end–stage renal disease; GCS, Glasgow coma scale; HR, heart rate; Hb, hemoglobin; IICP, increased intracranial pressure; MAP, mean blood pressure; MEDS, Mortality in Emergency Department Sepsis Score; MEWS, Modified Early Warning Score; NEWS, National Early Warning Score; PLT, platelet; qSOFA, quick Sequential Organ Failure Assessment; RAPS, Rapid Acute Physiology Score; REMS, Rapid Emergency Medicine Score; RR, respiratory rate; SBP, systolic blood pressure; WBC, white blood cells.

**Table 3 jpm-13-01358-t003:** Univariate Cox regression analyses for predisposing factors on clinical outcomes in 42 patients of cryptococcal fungemia.

Characteristics	Hazard Ratios	95% Confidence Interval	*p*-Value
Age (years)	1.03	(1.00–1.06)	0.023 *
Female	2.45	(1.11–5.42)	0.027 *
Focus of cryptococcosis			
CNS	0.36	(0.16–0.85)	0.013 *
Concomitant infection			
LRTI	2.59	(1.15–5.81)	0.021 *
Vital signs			
RR (bpm)	1.18	(1.06–1.32)	0.002 **
GCS	0.92	(0.84–0.97)	0.008 **
Laboratory data			
WBC (counts/uL)	1.00	(1.00–1.00)	0.028 *
PLT (×10^3^ counts/uL)	1.00	(0.99–1.00)	0.026 *
Crea (mg/dL)	1.17	(1.03–1.32)	0.016 *
Lactate (mg/dL)	1.03	(1.00–1.05)	0.004 **
Clinical management			
O_2_ use	4.74	(1.64–13.73)	0.004 **
Scoring systems			
REMS	1.18	(1.06–1.32)	0.003 **
RAPS	1.30	(1.11–1.53)	0.001 **
MEWS	1.37	(1.14–1.66)	0.001 **
MEWS with GCS	1.29	(1.12–1.48)	<0.001 **
MEDS	1.18	(1.08–1.28)	<0.001 **
NEWS	1.19	(1.09–1.30)	<0.001 **
qSOFA	2.11	(1.47–3.02)	<0.001 **

Cox regression analysis. * *p* < 0.05, ** *p* < 0.01, Statistically significant. BT, body temperature; CNS, central nervous system; Crea, Creatinine; DBP, diastolic blood pressure; DM, Diabetes Mellitus; ESRD, end–stage renal disease; GCS, Glasgow coma scale; HR, heart rate; Hb, hemoglobin; IICP, increased intracranial pressure; MAP, mean blood pressure; MEDS, Mortality in Emergency Department Sepsis Score; MEWS, Modified Early Warning Score; NEWS, National Early Warning Score; PLT, platelet; qSOFA, quick Sequential Organ Failure Assessment; RAPS, Rapid Acute Physiology Score; REMS, Rapid Emergency Medicine Score; RR, respiratory rate; SBP, systolic blood pressure; WBC, white blood cells.

**Table 4 jpm-13-01358-t004:** Univariate and multivariate Cox regression analyses for scoring systems on the in-hospital mortality rate in 42 patients of cryptococcal fungemia.

	Univariate			Multivariate		
Variables	HR	95% CI	*p*-Value	HR	95% CI	*p*-Value
MEDS	1.18	(1.08–1.28)	<0.001 **	1.20	(1.03–1.40)	0.018 *
NEWS	1.19	(1.09–1.30)	<0.001 **	1.03	(0.77–1.39)	0.813
MEWS with GCS	1.29	(1.12–1.48)	<0.001 **	1.09	(0.59–2.04)	0.766
MEWS	1.37	(1.14–1.66)	0.001 **	0.84	(0.40–1.75)	0.647
RAPS	1.30	(1.11–1.53)	0.001 **	1.24	(0.88–1.74)	0.211
REMS	1.18	(1.06–1.32)	0.003 **	0.84	(0.65–1.09)	0.196
qSOFA	2.11	(1.47–3.02)	<0.001 **	1.73	(0.61–4.88)	0.298

Cox regression analysis. * *p* < 0.05, ** *p* < 0.01, statistically significant. GCS, Glasgow coma scale; MEDS, Mortality in Emergency Department Sepsis Score; MEWS, Modified Early Warning Score; NEWS, National Early Warning Score; qSOFA, quick Sequential Organ Failure Assessment; RAPS, Rapid Acute Physiology Score; REMS, Rapid Emergency Medicine Score.

**Table 5 jpm-13-01358-t005:** The AUC of ROC, COP, sensitivity, specificity, PPV, NPV, accuracy, and SE of scoring systems to predict mortality risk.

Scores	AUC	COP	Sensitivity	Specificity	PPV	NPV	Accuracy	SE	*p*-Value
MEDS	0.905	4	93%	80%	89%	86%	88%	0.047	<0.001 **
NEWS	0.878	5	93%	87%	93%	87%	91%	0.069	<0.001 **
qSOFA	0.848	1	85%	80%	89%	75%	83%	0.064	<0.001 **
MEWS with GCS	0.846	3	89%	73%	86%	79%	83%	0.069	<0.001 **
REMS	0.846	8	70%	87%	91%	62%	76%	0.059	<0.001 **
RAPS	0.842	3	70%	100%	100%	65%	81%	0.061	<0.001 **
MEWS	0.833	3	93%	73%	86%	85%	86%	0.071	<0.001 **

** *p* < 0.01, Statistically significant. AUC, area under the curve; COP, cut-off point; GCS, Glasgow coma scale; MEDS, Mortality in Emergency Department Sepsis Score; MEWS, Modified Early Warning Score; NEWS, National Early Warning Score; NPV, negative predictive value; PPV, positive predictive value; qSOFA, quick Sequential Organ Failure Assessment; RAPS, Rapid Acute Physiology Score; REMS, Rapid Emergency Medicine Score; ROC, receiver operating characteristic curve; SE, standard error.

## Data Availability

Readers can access the data and material supporting the study’s conclusions by contacting Sung-Yuan Hu at song9168@pie.com.tw.

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
