# Peer review of "Predictive Performance of Scoring Systems for Mortality Risk in Patients with Cryptococcemia: An Observational Study"

_jpm, 2023, doi:10.3390/jpm13091358_

Round 1

Reviewer 1 Report

In this clinical data manuscript, authors attempted to collate retrospective data from single hospital to correlate or to get a robust scoring system to predict the outcome of the patients with cryptococcal fungemia. The authors do agree with the several limitations and the reviewer did not find a novel way of addressing the current issue. The sample size for each condition is pretty low and hard to interpret. It seems CNS involvement may not a primary factor of survival in this study. However, given the conditions that patients are undergoing, the survival rate rather dependent on other ongoing conditions such as other organs failure/condition involvement. Yet, such a report merit for publication. It would have been really a good report if authors collated data from the other hospital or included larger population size.

Author Response

Response to Reviewer 1:

Comments and Suggestions for Authors

In this clinical data manuscript, authors attempted to collate retrospective data from single hospital to correlate or to get a robust scoring system to predict the outcome of the patients with cryptococcal fungemia.

Response:

Thank you for the reviewer’s comment and understanding of the aim of our study to attempt to predict the clinical outcome of patients with cryptococcal fungemia by the scoring systems.

The authors do agree with the several limitations and the reviewer did not find a novel way of addressing the current issue. The sample size for each condition is pretty low and hard to interpret.

Response:

Thank you for the reviewer's efforts to strengthen our manuscript and point out the weaknesses of our study. As our study's retrospective character and limited case number, we would want further cooperation with other hospitals or a more extended data collection period in additional study design.

It seems CNS involvement may not a primary factor of survival in this study. However, given the conditions that patients are undergoing, the survival rate rather dependent on other ongoing conditions such as other organs failure/condition involvement. Yet, such a report merit for publication.

Response:

Thank you for the reviewer’s comment and appreciation of our manuscript and the result of our study. In our research, whether CNS involvement in the patients who received CSF examination is not a factor affecting survival rate but other scoring systems or patients’ conditions affecting the survival rate, which is the significant result in our study.

It would have been really a good report if authors collated data from the other hospital or included larger population size.

Response:

Thank you for the reviewer’s comment and suggestion to strengthen our manuscript and point out the limitations of our study for further study design.

The authors make a revised manuscript according to the reviewer’s comments and suggestion.

The correspondence

Sung-Yuan Hu, Department of Emergency Medicine, Taichung Veterans General Hospital, Taiwan.

Reviewer 2 Report

Thank you for submitting your draft titled "Predictive Performance of Scoring Systems for the Mortality Risk in Patients with Cryptococcemia: An Observational Study" . I appreciate your effort and  I am grateful to submit my comments and suggest changes to improve the paper. I request only minor clarifications in the remarks that I add in the draft in the attached .pdf file.

Author Response

Response to Reviewer 2:

Comments and Suggestions for Authors

Thank you for submitting your draft titled "Predictive Performance of Scoring Systems for the Mortality Risk in Patients with Cryptococcemia: An Observational Study". I appreciate your effort and I am grateful to submit my comments and suggest changes to improve the paper. I request only minor clarifications in the remarks that I add in the draft in the attached PDF file.

Response:

Thank you for the reviewer's comment and suggestion; our manuscript's detailed markings allow us to strengthen our study and correct the easily misunderstood sentence. Moreover, thank for the reviewer's appreciation of our manuscript and the effort of the study.

1. Was cryptococcal fungemia defined as growth of the fungus in blood cultures and/or presence of antigen? which antigen brand was used? Was titration performed? add this to the text and discuss.

Response:

Thanks for reviewer’s comments. In our manuscript, patients were included in study only if cryptococcus fungus was identified growth in blood culture.

The authors revised the manuscript as “We excluded patients only presenting the positive cryptococcal antigen without growth of cryptococcus in blood culture.” in the part of 2.1. Data Collection and Definition.  

2. Please describe this paragraph better. For a clinical site to be considered positive, how many positive samples in India ink were needed? What clinical sites are evaluated in this analysis?

Response:

Thank you for the reviewer’s suggestion and give us the chance to improve the paragraph. After reviewed our study population, all CNS infection (22 patients), respiratory infection (4 patients), urinary infection (2 patients), peritoneal infection (1 patient), and skin infection (1 patient) are defined as positive culture of Cryptococcus neoformans from specimens of the clinically sites (CSF, sputum/bronchial lavage, urine, ascites, and skin biopsy) in the part of 3. Result.

The authors avoid misunderstood, so we will modify this sentence to “We defined cryptococcosis as a positive culture of Cryptococcus neoformans yielded from the various specimens of the clinically involved sites, including cerebrospinal fluid (CSF), sputum/bronchial lavage, urine, ascites, and skin biopsy.” in the part of 2.1. Data Collection and Definition.

3. Please improve the understanding of the sentence.

Response:

Thank you for the reviewer’s suggestion to make our manuscript more understandable

The authors modified this sentence as “Our study identified no other cryptococcal species but Cryptococcus neoformans in patients with cryptococcal fungemia.” in the part of 3.3 Microbiology.

4. Italics

Response:

Thank you for the reviewer pointing out the error in our manuscript.

The authors revised the “italics” type of Cryptococcus neoformans in the part of 3.3 Microbiology.

5. Are these data from this article? If they are not, rephrase the sentence, for example: Recent work has shown or data from the literature report ......

Response:

Thank you for the reviewer’s suggestion. These data are from the published literature report.

The authors modified this sentence as “The published literature report showed a relatively higher growth rate in the planktonic type at 300C compared to 350C but better biofilm growth at 350C. ” in the part of Discussion.

6. Cryptococcemia was not rare 28% (12/42). I suggest modifying this sentence.

Response:

Thank you for the reviewer’s the suggestion and comment to give us the chance to clarify this easily misunderstanding sentence. In this sentence, we try to express that cryptococcemia is a rare disease with therefore lack of awareness in patient initial presented in emergency department, and also the long inoculated time of cryptococcus species growth, which make it difficult to conduct a prospective study and standardized treatment protocol for cryptococcemia patients.

The authors modified this sentence as “Fourth, cryptococcemia is a rare disease with, therefore, lack of awareness in patients initially presented in the ED, and also the long inoculation time of cryptococcus species growth, which makes it challenging to conduct a prospective study and standardized treatment protocol for patients with cryptococcemia”

The authors make a revised manuscript according to the reviewer’s comments and suggestion.

The correspondence

Sung-Yuan Hu, Department of Emergency Medicine, Taichung Veterans General Hospital, Taiwan.
